# Prevalence of *Staphylococcus aureus* and *mec-A* Cassette in the Throat of Non-Hospitalized Individuals Randomly Selected in Central Italy

**DOI:** 10.3390/antibiotics11070949

**Published:** 2022-07-14

**Authors:** Luca Scapoli, Annalisa Palmieri, Agnese Pellati, Francesco Carinci, Dorina Lauritano, Claudio Arcuri, Luigi Baggi, Roberto Gatto, Marcella Martinelli

**Affiliations:** 1Department of Experimental, Diagnostic and Specialty Medicine, University of Bologna, 40138 Bologna, Italy; luca.scapoli2@unibo.it (L.S.); marcella.martinelli@unibo.it (M.M.); 2Department of Translational Medicine, University of Ferrara, 44121 Ferrara, Italy; agnese.pellati@unife.it (A.P.); francesco.carinci@unife.it (F.C.); dorina.lauritano@unife.it (D.L.); 3Department of Clinical Sciences and Translational Medicine, University of Rome “Tor Vergata”, 00113 Rome, Italy; claudio.arcuri@uniroma2.it (C.A.); luigi.baggi@uniroma2.it (L.B.); 4Department of Life, Health and Environmental Sciences, School of Dentistry, University of L’Aquila, 67100 L’Aquila, Italy; roberto.gatto@univaq.it

**Keywords:** methicillin-resistant *Staphylococcus aureus* (MRSA), antibiotics, *mec-A* gene, real-time PCR, non-hospitalized population, Central Italy

## Abstract

Methicillin-resistant *Staphylococcus aureus* (MRSA) is a cause of life-threatening infections that are difficult to treat because of resistance to several antibiotics. Most documented MRSA infections are acquired nosocomially or among community with frequent contact with health facilities. However, an increasing attention to community acquired MRSA strains appears justified. A population of Central Italy was investigated for the presence of *S. aureus* and for the methicillin-resistance determinant *mec-A* gene. Exclusion was due to systemic diseases, pathologies or therapies inducing systemic immunosuppression, facial trauma or poor oral hygiene. Throat swabs obtained from 861 randomly selected participants were tested for the presence of DNA sequences of *S. aureus* and the *mec-A* gene by real-time PCR. The DNA of *S. aureus* was detected in 199 specimens (23.1%), while the *mec-A* gene was detected in 27 samples (3.1%). The prevalence of patients carrying methicillin-resistant strains was higher in younger and older strata. The prevalence of *mec-A* among *S. aureus* positive samples was 7.5%. Our data confirm that *S. aureus* and methicillin-resistant strains are common in the throat of the general population of Central Italy. Although the PCR methods used in this study are different from traditional culture-based approaches, the observed prevalence was consistent to those observed in Italians and other populations. Considering that carriers have a higher risk to develop post surgically life-threatening infections, it is worth evaluating a preventive approach based on rapid PCR screening of incoming patients to reduce the risk of developing health-care-associated infections.

## 1. Introduction

Staphylococci frequently asymptomatically colonize the skin and mucous membranes of humans and animals. Among several species of staphylococci, *Staphylococcus aureus* shows the highest pathogenic potential. Indeed, in the bloodstream or in the internal tissues, this bacterium may cause life-threatening infections [1]. *S. aureus* is a leading cause of bacteraemia and infective endocarditis; however, infections commonly involve the skin, soft tissue, bone, joints and is commonly associated with indwelling catheters and prosthetic devices [2].

A fundamental biological property of *S. aureus* is its ability to thrive on intact epithelia causing no symptoms. The most frequent carriage site for *S. aureus* are nares, skin, perineum and pharynx. Interestingly, a comparison of the different sampling sites revealed higher carriage rates in the throat than in the nares [3]. Roughly 30% of humans are carriers of *S. aureus* as part of their normal flora [4]. Several investigations reported both bacterial and human determinants associated with *S. aureus* colonization [5].

An increasing bulk of evidence supports that carriers are more susceptible to infections [6]. Nosocomial *S. aureus* bacteraemia was three-times more frequent in *S. aureus* carriers than in non-carriers; moreover, 80% of bacteraemic *S. aureus* isolates were endogenous, since isolates obtained from blood were clonally identical to the isolates obtained from the nares before bacteraemia [6]. This suggests that bacteria colonizing the mucosae could be the source of blood infection and that asymptomatic carriers could be at a higher risk of infection.

Despite the use of antibiotics and improved health care, *S. aureus* remains one of the major causes of hospital-related infections [7]. *S. aureus* is virtually sensitive to all antibiotics; however, the bacterium can easily acquire resistance against all classes of antibiotics through mutation of a bacterial gene or by horizontal transfer of a resistance gene from another bacterium. The most important genetic element that *S. aureus* could horizontally acquire is the staphylococcal cassette chromosome mec (SCCmec) because it carries the *mec-A* gene, a central determinant for broad-spectrum antibiotic resistance [8]. The *mec-A* encodes the penicillin-binding protein 2a conferring resistance to methicillin and β-lactam antibiotics, including penicillins, cephalosporins and carbapenems.

The origin of SCCmec is unknown; however, data support the hypothesis that *mec-A* in all staphylococci descended from one common ancestor, possibly the *Staphylococcus sciuri* and then spread by horizontal transfer among different staphylococcal species [9,10].

The first reports describing isolation of a *S. aureus* strain that was resistant to methicillin from clinical specimens were published in 1961, readily after methicillin was introduced [11]. Methicillin-resistant *S. aureus* (MRSA) strains are now a leading cause of health care-associated infections worldwide and have emerged as a major cause of community-associated infections [12].

*Staphylococcus epidermidis* and other coagulase-negative staphylococci are less pathogenic than *S. aureus*; however, they are emerging as causative agents in nosocomial infections because of their ability to produce several adhesion molecules and generate biofilms. A recent paper reported that periprosthetic joint infections by methicillin-resistant *S. epidermidis* are much more difficult to eradicate than *S. aureus* [13].

There are few reports giving an account of the prevalence of *S. aureus* or MRSA in the non-hospitalized Italian population [14,15,16]. With respect to hospitalized patients, interesting data can be retrieved from a bacteriaemia national surveillance program (Report of antibiotic resistance by Istituto Superiore di Sanità https://www.epicentro.iss.it/antibiotico-resistenza/ar-iss, accessed on 15 June 2022). *S. aureus* has been isolated from 20% of blood samples from patients with bacteraemia; about of one-third of the samples tested positive for MRSA. A higher MRSA prevalence (>40%) was observed in regions of Central Italy, where the subjects of the present study were enrolled.

Knowledge of antibiotic resistance patterns in hospitalized or in other exposed populations represents a major requirement in the management of MRSA infection that has been widely investigated. However, it has become increasingly urgent to evaluate additional potential reservoirs, such as the general population. In addition, considering that SCCmec can be transferred among staphylococci, it is important to evaluate its local prevalence. To these aims, the present investigation explored the prevalence of *mec-A* and *S. aureus* sequences in throat swabs from randomly selected people from Central Italy.

## 2. Results

In this research, a sample of randomly selected participants were investigated for the presence of *S. aureus* and *mec-A* gene sequences in throat swabs. The laboratory received and processed for DNA analysis a total of 892 swabs from volunteers who signed an informed consent. Two samples were excluded from further analyses because the total bacterial load was under the established threshold of 20,000 units. An additional 29 samples were excluded because of missing or incomplete data in the accompanied questionnaire.

The remaining 861 samples were tested for the presence of *S. aureus*, *mec-A* gene and *S. epidermidis*. The age and sex distribution of patients was shown in the pyramid plot (Figure 1); it revealed an overrepresentation of the age class between 20 and 30 years and an overrepresentation of female patients. Specifically, 39% of patients were male, 34% were smokers, and 37% consumed alcoholic beverages (Table 1).

The DNA of *S. aureus* was detected by real-time PCR in 199 specimens, while the *mec-A* gene was detected in 27 samples. The calculated prevalences were 23.1% (95% C.I. 20.4–26.0) and 3.1% (95% C.I. 2.1–4.5), respectively. The *S. epidermidis* was detected in almost every sample, indeed the prevalence was 99.1% (95% C.I. 98.2–99.5). Figure 2 reports the distribution of age-specific prevalence. The highest prevalence of *S. aureus* was observed in younger people with a gradual decline with increasing age. The prevalence of *mec-A* instead appeared above the average among youngest (12% (95% C.I. 3.5–36)) and oldest (6.6% (95% C.I. 3.1–14)) people.

The prevalence of *mec-A* among *S. aureus* positive samples was 7.5% (95% C.I. 4.6–12) (Table 2).

Some of the *mec-A* positive samples were *S. aureus* negative, although there is a strong association between the occurrence of *S. aureus* and *mec-A* cassette (*p* value = 0.00004) with an OR = 4.4 (95% C.I. 2.0–9.6). In addition, a strong correlation was observed between the amount of *mec-A* gene and *S. aureus* genome in double positive samples (Pearson’s r = 1; *p* value < 0.001). These data suggest that specimens positive for either *mec-A* and *S. aureus* were likely due to MRSA strains. The correlation between *mec-A* and *S. epidermidis* was not significant in the whole sample (Pearson’s r = −0.02; *p* value = 0.92) but was borderline in the *S. aureus* negative subgroup (Pearson’s r = 0.5; *p* value = 0.08), suggesting that *S. epidermidis* could be a reservoir of *mec-A* in specimens lacking *S. aureus*.

The occurrence of *S. aureus* was not associated with the sex of the subjects but was slightly associated with a smoking habit (*p* value = 0.04; OR = 1.4 (95% C.I. 1.0–2.0)) and with alcohol consumption (*p* value < 0.05; OR = 1.4 (95% C.I. 1.0–1.9)). The occurrence of *mec-A* was not associated with sex, smoking habits or alcohol consumption (Table 3).

## 3. Discussion

*S. aureus* is a Gram-positive bacterium that colonizes the skin and mucosae in about 25–30% of healthy people. However, *S. aureus* can cause a range of diseases if it crosses the epithelial barrier and accesses underlying tissues. This major human pathogen is a common cause of systemic infections, such as infective endocarditis, osteomyelitis, epiglottitis and sinus infections amongst others. Such infections can be life-threatening when uncured or in case of multi-drug resistance that can be acquired through mutational changes or via horizontal gene transfer by other bacteria [8]. In this investigation, we evaluated the diffusion of specific DNA sequences of the *mec-A* gene conferring methicillin and β-lactam resistance to *S. aureus* and other staphylococci in 861 randomly selected non-hospitalized individuals.

The presence of *mec-A*, *S. aureus* and *S. epidermidis* sequences were ascertained by specific real-time PCR assays. The prevalence of bacteria carrying the *mec-A* gene was 3.1%. Our data indicate that *S. aureus* is a main carrier of the *mec-A* gene in the investigated population. Indeed, the occurrence of *mec-A* sequences and *S. aureus* sequences were strongly associated (*p* value = 0.00004; OR = 4.4), and the amounts of the two targets were highly correlated (Pearson’s r = 1; *p* value < 0.001). In other words, there is a high chance that the 15 samples containing both *mec-A* and *S. aureus* sequences did actually include MRSA strains.

In this instance, the MRSA represented the 7.5% of the *S. aureus* positive samples. As expected, the burden of *mec-A* was not completely attributable to *S. aureus*, indeed 12 out of 27 of *mec-A* positive samples did not contain *S. aureus* sequences. This clearly indicates that other bacteria, most likely coagulase negative staphylococci, could represent a reservoir for horizontal transfer of the SCCmec to *S. aureus*.

A borderline correlation between amounts of *mec-A* and *S. epidermidis* sequence copies suggests that *S. epidermidis* could represent a candidate carrier of SCCmec. In a survey among college students, it was found that the prevalence of methicillin resistance of *S. epidermidis* was double that of *S. aureus* [17]. Wielders and coworkers [18] reported a possible horizontal in vivo transfer of *mec-A* between *S. epidermidis* and *S. aureus* during antibiotic treatment, and a sporadic MRSA emerged de novo.

In our sample, the overall prevalence of *S. aureus* was 23.1%, with higher levels observed among the younger, i.e., 38% in the age group of 0–10 years. These levels agree with *S. aureus* nasal carriers previously observed in scholars and adults by other authors in the Italian population [19]. Interestingly, a level as low as 10% was observed in people >60 years. The age-specific prevalence of *mec-A* was higher among subjects <10 and >60 years old and lower in middle age. This bimodal trend was similar to the nose MRSA prevalence in USA and in Pakistan general populations [20,21]. A possible explanation could be positive selection in age groups with higher level of antibiotic prescription [22].

In each part of the world, nosocomial MRSA infections are considered a major public-health threat. A reduction of MRSA transmission in hospitals has been obtained by adopting special precaution protocols with patients and health care personnel identified as MRSA carriers by preventive screening. The published cost-benefit analysis agrees that the costs of screening and anti-diffusion precautions are much lower than the cost of caring for patients who become infected with MRSA [12].

In this investigation, we used a PCR-based method to identify two risk factors independently, i.e., the presence of *S. aureus* and the presence of unidentified *mec-A*-based methicillin-resistant strains. The disadvantage of PCR, compared with the culture-based methods used in clinical microbiology laboratories, is the inability to clearly define the presence of MRSA; indeed, *mec-A* could also be carried by other staphylococci.

On the other hand, the PCR method is far more rapid, a few hours instead of 24–48 h. Moreover, it helps to reveal other methicillin-resistant staphylococci, which could be either a direct health care danger or could represent a reservoir of SCCmec that could be horizontally transferred to *S. aureus*. Thus, with a view to prehospital screening for the reduction of infection spread by antibiotic-resistant bacteria, the PCR approach has attractive advantages that make it preferable.

Our data confirm that *S. aureus* and most likely MRSA are common in the throat of the general population of Central Italy with comparable prevalence to observations in other populations. The overall prevalence of *mec-A* was even higher, indicating that other methicillin-resistant bacteria are common in the population and could represent a direct source of infection and a reservoir of SCCmec to develop new strains of MRSA. Considering that carriers have a higher risk of developing life-threatening infections, it is worth evaluating a preventive approach to reduce the risk of developing post-health-care infection.

## 4. Materials and Methods

### 4.1. Sample Collection

The human subjects included in the present study were randomly selected among dental clinics of the University of Tor Vergata (Italy) and from the University of L’Aquila (Italy) between December 2017 and March 2019. Exclusion criteria were the presence of systemic diseases (such as diabetes, heart failure, hypertension, renal failure, hepatic failure and respiratory failure), pathologies inducing systemic immunosuppression or the use of immunomodulatory or immunosuppressive drugs. Patients with facial trauma, those who had undergone radio and/or chemotherapy or patients with poor oral hygiene were also excluded. This sample cohort was previously examined for Human Papilloma Virus prevalence in the oropharynx of healthy individuals [23].

Patients who had given their consent were included in the study. Informed consent was signed by the patient or parents (for children) prior to sample collection. This study was approved by the L’Aquila ethics committee (approval number 26/2017).

A sample of the oral flora was collected with a throat swab from the surface of tonsils and oropharynx. Swabs were immediately placed in a test tube containing silica gel capsules, which rapidly dried the sample and preserved the integrity of the biological material, which was then stored at 4 °C until processing.

### 4.2. Molecular Analysis

Specimens were processed to extract the total DNA with two consecutive incubations in a lysis buffer with lysozyme and proteinase K in order to ensure indiscriminate cell lysis. Once extracted, DNA was purified on silica membranes using the QIAxtractor automatic instrument and a dedicated QIAcube HT purification kit (Qiagen, Hilden, Germany).

The absolute quantification assay of each specific target was performed by real-time PCR using the ABI PRISM 7500 thermal cycler (Applied Biosystems, Foster City, CA, USA). Each hydrolysis probe reaction was performed in 20 µL containing 10 µL of 2× qPCRBIO Probe Mix Lo-ROX, 100 ng of DNA purified from samples, 200 nM of each primer and 100 nM probe of each assay. The amplification profile started with 10′ at 95 °C to activate the polymerase and was followed by a two-step amplification of 15′′ at 95 °C and 60′′ at 57 °C for 40 cycles. Plasmids containing synthetic DNA target sequences (Eurofin MWG Operon) were used as standards for quantitative analysis. Standard curves, i.e., threshold cycle values against the log of the copy number, were constructed with the serial dilutions of plasmids ranging from 10 to 10,000,000 copies to check the amplification efficiency and for the quantification of the targets in each sample.

Three real-time PCR runs were performed for each sample. The first reaction was to evaluate the total amount of bacteria using two degenerate primers and a single probe matching a highly conserved 16 S rRNA gene sequence. This represented a quality control step to ensure that each step of the process (the sampling, conservation, DNA extraction and DNA purification) was successful. Highly specific primer-probe sets were designed with the Primer-BLAST tool [24]. The primer quality was further checked with MFEprimer v3.0 software, which also helped to set the multiplex PCR assays [25]. A multiplex reaction detected the *mec-A* gene and a specific *S. aureus* sequence of the *nuc* gene. A simplex reaction detected the *S. epidermidis* by targeting the 16 S rRNA gene sequence. Oligonucleotides were synthesized by Biomers.net (Ulm, Germany), and their sequences are reported in Table 4.

### 4.3. Statistical Analysis

Descriptive statistics and data analysis were performed using SPSS software v.25 (IBM, New York, NY, USA). Associations between variables were analyzed using Pearson’s chi square test of two-way contingency tables; two sided *p* values > 0.05 were considered significant. Relationships between variables were evaluated using the odds ratio (OR). The confidence limits for prevalence were calculated using the binomial proportions tool of the OpenEpi web site (www.openepi.com, accessed on 2 May 2022).

## Figures and Tables

**Figure 1 antibiotics-11-00949-f001:**
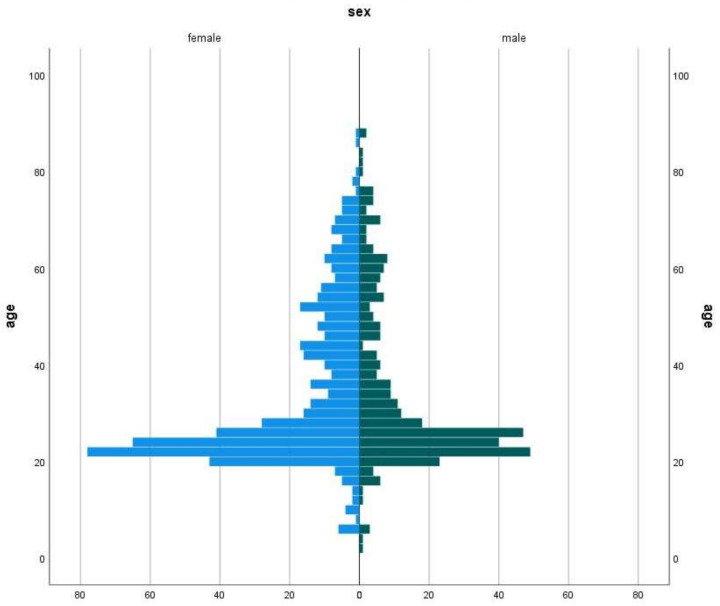
Age and sex distribution of study participants.

**Figure 2 antibiotics-11-00949-f002:**
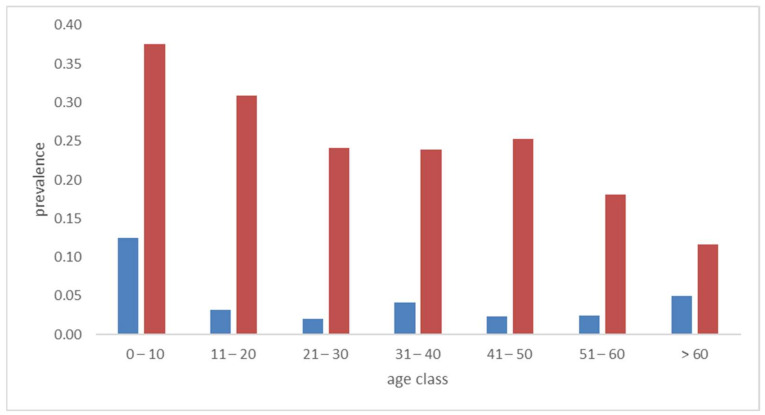
Prevalence of *S. aureus* (red bars) and *mec-A* gene (blue bars) in the different age class.

**Table 1 antibiotics-11-00949-t001:** Descriptive data of participants of the study.

	No		Yes	
Male	527	61%	334	39%
Smoking	568	66%	293	34%
Alcohol	537	62%	235	37%

**Table 2 antibiotics-11-00949-t002:** Contingency table showing the occurrence of *S. aureus* and *mec-A* gene.

		*mec-A*		Total
		(−)	(+)	
*S. aureus*	(−)	650	12	662
	(+)	184	15	199
Total		834	27	861

**Table 3 antibiotics-11-00949-t003:** Occurrence of *S. aureus* and *mec-A* stratified by sex, smoking habits and alcohol consumption.

		*S. aureus*		*p* Value	OR (95% C.I.)
		(−)	(+)		
Sex	Male	257	77	0.97	1.0 (0.73–1.4)
	Female	405	122		
Smoking	(−)	449	119	0.04	1.4 (1.0–2.0)
	(+)	213	80		
Alcohol	(−)	425	112	<0.05	1.4 (1.0–1.9)
	(+)	238	87		
		** *mec-A* **		***p* value**	**OR (95% C.I.)**
		(−)	(+)		
Sex	Male	323	11	0.83	0.92 (0.42–2.0)
	Female	511	16		
Smoking	(−)	548	20	0.37	0.67 (0.28–1.6)
	(+)	286	7		
Alcohol	(−)	520	17	0.94	0.97 (0.44–2.1)
	(+)	315	10		

**Table 4 antibiotics-11-00949-t004:** Oligonucleotide sequences for real-time PCR.

	Primer (5′–3′)	Probe (5′–3′)
*S. aureus nuc*	F-cacctgaaacaaagcatcctaaa	CY5-tggtcctgaagcaagtgcatttacgaaa
	R-gacctttgtcaaactcgacttca	
SCCmec/*mec-A*	F-agttagattgggatcatagcgtca	JOE-ccaggaatgcagaaagaccaaagcataca
	R-gccaattccacattgtttcg	
*S. epidermidis*	F-gaaccttaccaaatcttgacatcctc	FAM-ccctctagagatagagttttccccttcggg
	R-tgcaccacctgtcactctgtc	
Total bacteria	F-acgcgargaccttacchr	FAM-cacgagctgacgacarccatgca
	R-gsacttaasccracatctca	

## Data Availability

The data used to support the findings of this study are available from the corresponding author upon request.

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
