# Peer review of "Prevalence of Staphylococcus aureus and mec-A Cassette in the Throat of Non-Hospitalized Individuals Randomly Selected in Central Italy"

_antibiotics, 2022, doi:10.3390/antibiotics11070949_

Round 1

Reviewer 1 Report

Title

Staphylococcus aureus and mec-A cassette throat prevalence in a random sample of non-hospitalized population of Central Italy

Abstract

Methicillin-resistant Staphylococcus aureus (MRSA) is a cause of life-threatening infections that are difficult to treat because of resistance to several antibiotics. Most documented MRSA infec-tions were acquired nosocomially or among community with frequent contact with health facil-ities. However, an increasing attention to community acquired MRSA strains appears justified. A population of Central Italy was investigated for the presence of S. aureus and for the methicil-lin-resistance determinant mec-A gene. Throat swabs obtained from 861 randomly selected par-ticipants were tested for the presence of DNA sequences of S. aureus and mec-A gene by re-al-time PCR. The DNA of S. aureus was detected in 184 specimens (23.1%), while mec-A gene was detected in 27 samples (3.1%). The prevalence of patients carrying methicillin-resistant strains is higher in younger and older strata. The prevalence of mec-A among S. aureus positive samples was 7.5%. Our data confirm that S. aureus and methicillin-resistant strains are common in the throat of general population of Central Italy. Although the PCR methods used in this study was different from traditional culture-based approaches, the observed prevalence was consistent to those ob-served in Italians and other populations. Considering that carriers have higher risk to develop post surgically life-threatening infections it is worth evaluating a preventive approach based on a rapid PCR screening of incoming patients to reduce the risk of developing health care associate infection.

Comment: inclusion and exclusion criteria should be included

Main issues: Prevalence and incidence are frequently confused. Prevalence refers to proportion of persons who have a condition at or during a particular time period, whereas incidence refers to the proportion or rate of persons who develop a condition during a particular time period. So prevalence and incidence are similar, but prevalence includes new and pre-existing cases whereas incidence includes new cases only. The key difference is in their numerators.

Numerator of incidence = new cases that occurred during a given time period

Numerator of prevalence = all cases present during a given time period

The numerator of an incidence proportion or rate consists only of persons whose illness began during the specified interval. The numerator for prevalence includes all persons ill from a specified cause during the specified interval regardless of when the illness began. It includes not only new cases, but also preexisting cases representing persons who remained ill during some portion of the specified interval.

Prevalence is based on both incidence and duration of illness. High prevalence of a disease within a population might reflect high incidence or prolonged survival without cure or both. Conversely, low prevalence might indicate low incidence, a rapidly fatal process, or rapid recovery.

Prevalence rather than incidence is often measured for chronic diseases such as diabetes or osteoarthritis which have long duration and dates of onset that are difficult to pinpoint.

Typo: The remining 861 samples were tested for the presence of

Smking

The human subjects included in the present study were randomly selected among  dental clinics of the University of Tor Vergata (Italy) and from the University of L’Aquila 207 (Italy

-comment: pls explain how randomization was conducted

Our data confirm that S. aureus and most likely MRSA are common in the throat of 197 general population of Central Italy with comparable prevalence to those observed in other 198 populations. The overall prevalence of mec-A is even higher, indicating that other methi-199 cillin-resistant bacteria are common in the population and could represent a direct source 200 of infection and a reservoir of SCCmec to develop new strains of MRSA. Considering that 201 carriers have higher risk to develop life-threatening infections it is worth evaluating a pre-202 ventive approach to reduce the risk of developing post health care infection.

Comment: the results seems overinflated, pls check. 

Author Response

Response to Reviewer 1 Comments

Point 1: inclusion and exclusion criteria should be included (in the abstract)

Response 1: In response to the Reviewer’s comment, we added this sentence in the abstract: “Exclusion was because of systemic diseases, pathologies or therapies inducing systemic immunosuppression, facial trauma, or poor oral hygiene.”

We believe that there is no need of inclusion criteria because of random selection.

A sentence stated that sample study was randomly selected, this means that there were no inclusion criteria (see also the response at Point 4)

Point 2: Main issues: Prevalence and incidence are frequently confused. Prevalence refers to proportion of persons who have a condition at or during a particular time period, whereas incidence refers to the proportion or rate of persons who develop a condition during a particular time period. So prevalence and incidence are similar, but prevalence includes new and pre-existing cases whereas incidence includes new cases only. The key difference is in their numerators.

Numerator of incidence = new cases that occurred during a given time period

Numerator of prevalence = all cases present during a given time period

The numerator of an incidence proportion or rate consists only of persons whose illness began during the specified interval. The numerator for prevalence includes all persons ill from a specified cause during the specified interval regardless of when the illness began. It includes not only new cases, but also preexisting cases representing persons who remained ill during some portion of the specified interval.

Prevalence is based on both incidence and duration of illness. High prevalence of a disease within a population might reflect high incidence or prolonged survival without cure or both. Conversely, low prevalence might indicate low incidence, a rapidly fatal process, or rapid recovery.

Prevalence rather than incidence is often measured for chronic diseases such as diabetes or osteoarthritis which have long duration and dates of onset that are difficult to pinpoint.

Response 2: We completely agree with the Reviewer’s comment. For this reason, in the original manuscript, we always used the term prevalence. No changes were made in the revised manuscript.

Point 3: Typo: The remining 861 samples were tested for the presence of

Smking

Response 3: Typos have been corrected.

Point 4: The human subjects included in the present study were randomly selected among dental clinics of the University of Tor Vergata (Italy) and from the University of L’Aquila (Italy

-comment: pls explain how randomization was conducted

Response 4: Random selection and random assignment are commonly confused or used interchangeably, though the terms refer to different processes. Random selection refers to how study participants are selected from the population for inclusion in the study. Random assignment is an aspect of experimental design in which study participants are assigned to the treatment or control group using a randomization procedure that must be reported in methods.

In the present investigation we performed a random selection of the participants, a procedure that does not need description because it is not guided by any specific method by definition. Consequently, no additional information has been added to the edited manuscript.

Point 5: Our data confirm that S. aureus and most likely MRSA are common in the throat of general population of Central Italy with comparable prevalence to those observed in other populations. The overall prevalence of mec-A is even higher, indicating that other methicillin-resistant bacteria are common in the population and could represent a direct source of infection and a reservoir of SCCmec to develop new strains of MRSA. Considering that carriers have higher risk to develop life-threatening infections it is worth evaluating a preventive approach to reduce the risk of developing post health care infection.

Comment: the results seems overinflated, pls check.

Response 5: After careful reading and collegial discussion, authors feel that this sentence is adequate to close the Discussion section. Terms like “most likely”, “could represent”, and “is worth evaluating” give the appropriate level of uncertainness to some of the issues that were deeply discussed above, in the same section. No changes in this part have been made in the revised manuscript.

Reviewer 2 Report

The present study dealt with an interesting topic and in general is well organized! While reading, a few points should be considered:

- A suggestion to improve / modify the title, because from that the reader thinks he will read a large study, but in reality, it seems like a "short communication" in what interests the authors to communicate. The authors have used "Central Italy" and this region is large enough to be studied, and this creates confusion here. The authors considered only two dental clinics of the two universities, and a small sample (892 tampons). From this, the reader may have many questions, because it is not all of Central Italy; for this the referee suggests reformulating the article. It is too short to be considered an "article", where references are few.

- Starting with the abstract and ending with the last sentence of the manuscript, the authors should express in italics all the names of the bacteria, such as (S. aureusS. epidermidis and S. sciuri), and the words “in vivo, de novo”.

- In line 75 enter first the full name of the bacteria and then its acronym.

- The sentence in lines 80-84 needs a contextual improvement.

- In table 1 correct “Smoking”, line 110. Even in table 2, S. aureus needs to be expressed in italics.

- So, the sentence in lines 197-199 needs revision for Central Italy, where young people predominate, i.e., 38% in the 0-10 year age group…

- In Table 3, please enter the abbreviations of the bacteria and not their full names as shown.

- The subsection of statistics and data analysis should be created. In addition, sections of results and materials and methods should be organized into subsections to better understand the main messages of the study.

Author Response

Response to reviewer 2 comments

Point 1: The present study dealt with an interesting topic and in general is well organized!

Response 1: Authors would like to thank the Reviewer for this comment.

Point 2: While reading, a few points should be considered:

- A suggestion to improve / modify the title, because from that the reader thinks he will read a large study, but in reality, it seems like a "short communication" in what interests the authors to communicate. The authors have used "Central Italy" and this region is large enough to be studied, and this creates confusion here. The authors considered only two dental clinics of the two universities, and a small sample (892 tampons). From this, the reader may have many questions, because it is not all of Central Italy; for this the referee suggests reformulating the article. It is too short to be considered an "article", where references are few.

Response 2: The Authors have structured the manuscript as an Article since it is an original research paper, but if the Editor feels that another format would better fit the journal policies, authors are ready to accomplish her/his request.

The title has been revised on request of Reviewer 4: “Prevalence of Staphylococcus aureus and mec-A cassette in the throat of non-hospitalized individuals randomly selected in Central Italy“. Authors believe that the revised title would fit the Reviewer 2 request, too. Indeed, it is now clearer that sample was recruited in Central Italy, but because of random selection it does not aspire to represent the entire population of Central Italy. Moreover, the two University clinics where patients were enrolled do not treat patients only from Rome and L'Aquila, but from large areas surrounding the cities.

Point 3: - Starting with the abstract and ending with the last sentence of the manuscript, the authors should express in italics all the names of the bacteria, such as (S. aureus, S. epidermidis and S. sciuri), and the words “in vivo, de novo”.

Response 3: Authors agree with the Reviewer. Italics formatting was lost at some stage in the original manuscript uploading process. The revised version has been corrected.

Point 4: - In line 75 enter first the full name of the bacteria and then its acronym.

Response 4: Manuscript has been edited according to the Reviewer’s suggestion.

Point 5: - The sentence in lines 80-84 needs a contextual improvement.

Response 5: Based on the Reviewer’s suggestion, the paragraph has been improved as follows: “There are few reports giving an account of the prevalence of S. aureus or MRSA in non-hospitalized Italian population (PMID: 35526042 PMID: 33680341 PMID: 23731504). With respect to hospitalized patients, interesting data can be retrieved from a bacteriaemia national surveillance program (Report of antibiotic resistance by Istituto Superiore di Sanità https://www.epicentro.iss.it/antibiotico-resistenza/ar-iss). S. aureus has been isolated from 20% of blood samples from patients with bacteraemia; about of one-third of the samples tested positive for MRSA. To note that higher MRSA prevalence (> 40%) was observed in regions of Central Italy, where subjects of the present study were enrolled.”

Point 6: - In table 1 correct “Smoking”, line 110. Even in table 2, S. aureus needs to be expressed in italics.

- So, the sentence in lines 197-199 needs revision for Central Italy, where young people predominate, i.e., 38% in the 0-10 year age group…

- In Table 3, please enter the abbreviations of the bacteria and not their full names as shown.

Response 6: The Authors thank the Reviewer for improvement suggestions. Revisions have been made accordingly (now Table 4).

Point 7: - The subsection of statistics and data analysis should be created. In addition, sections of results and materials and methods should be organized into subsections to better understand the main messages of the study.

Response 7: The revised manuscript include a Materials and Methods section organized in subsections. A specific subsection for statistical details was added:

“Statistical analysis

Descriptive statistics and data analysis were performed using SPSS software v.25 (IBM, New York, NY, USA). Association between variables were analyzed by Pearson’s chi square test of two-way contingency tables; two sided P values > 0.05 were considered significant. Relationship between variables were evaluated by the odds ratio (OR). Confidence limits for prevalence were calculated by the binomial proportions tool of the OpenEpi web site (www.openepi.com).”

Authors believe that subsections would not improve readability of the Results section.

Reviewer 3 Report

1. The abstract is not structured. This could only be the case with a literature review, but this is not the case.

2. The whole structure is confused and the accepted sequence is not followed! It starts with an introduction, material and methods, results, discussion, discussion and used literature. Thanksgiving, conflict of interest sectors can be added. Please the authors to arrange the article according to requirements!

3. In clinical cases, it is good to mention whether an informed consent has been signed for the experiment, as well as whether patients agree to publish their results, whether the mine is an ethics committee for working with people… This part is performed by the authors.

4. The Ethics Committee passed in 2017. This suggests other similar studies. The authors could mention confirming or rejecting their preliminary results, which should not be considered an unacceptable self-citation.

Author Response

Response to Reviewer 3 comments

Point 1: The abstract is not structured. This could only be the case with a literature review, but this is not the case.

Response 1: The authors adhered to the guidelines imposed by the journal itself, that stated:

Abstract: The abstract should be a total of about 200 words maximum. The abstract should be a single paragraph and should follow the style of structured abstracts, but without headings: 1) Background: Place the question addressed in a broad context and highlight the purpose of the study; 2) Methods: Describe briefly the main methods or treatments applied. Include any relevant preregistration numbers, and species and strains of any animals used. 3) Results: Summarize the article's main findings; and 4) Conclusion: Indicate the main conclusions or interpretations.

Point 2: The whole structure is confused and the accepted sequence is not followed! It starts with an introduction, material and methods, results, discussion, discussion and used literature. Thanksgiving, conflict of interest sectors can be added. Please the authors to arrange the article according to requirements!

Response 2: The original manuscript complies with the guidelines imposed by the Journal itself, as below reported:

  • Research manuscripts should comprise:
    • Front matter: Title, Author list, Affiliations, Abstract, Keywords
    • Research manuscript sections: Introduction, Results, Discussion, Materials and Methods, Conclusions (optional).
    • Back matter: Supplementary Materials, Acknowledgments, Author Contributions, Conflicts of Interest, References.

For this reason, no changes were made to the structure of the manuscript.

Point 3: In clinical cases, it is good to mention whether an informed consent has been signed for the experiment, as well as whether patients agree to publish their results, whether the mine is an ethics committee for working with people… This part is performed by the authors.

Response 3: In lines 229-231 Authors mentioned that only patients who signed an informed consent were included in the study, and reported the reference of the study approval:

“Patients who had given their consent were included in the study. Informed consent was signed by the patient or parents (for children) prior to sample collection. This study was approved by the L’Aquila ethics committee (approval number 26/2017).”

Point 4: The Ethics Committee passed in 2017. This suggests other similar studies. The authors could mention confirming or rejecting their preliminary results, which should not be considered an unacceptable self-citation.

Response 4: Another recently published investigation about Human Papilloma Virus was performed and using the same sample study. A citation was added in the Materials and Methods section: “This sample cohort was previously examined for Human Papilloma Virus prevalence in the oropharynx of healthy individuals (PMID: (Palmieri, Lauritano et al. 2022)).”.

The previous and the present investigations were delayed because of the SARS-CoV-2 pandemic.

The present manuscript was not previously submitted to any other journal.

Reviewer 4 Report

The study aims to evaluate the prevalence of S. aureus and MRSA mec-A gene in the throat swabs from randomly selected Non-hospitalized population from Central Italy. New data generated seem to be just adequate. 

Some of the suggestions are 

Italicize the bacterial names through out the manuscript.

Title: Suggest to re-frame the title. Avoid using bacterial name at the start of title.

Abstract: Prevalence of S. aureus mentioned as 23.1% (Please cross check, should be 21.3%).

Introduction: Line 75. S epidermidis and other coagulase-negative....(Write full bacterial name at its first appearance in the text)

Results: Table 1. Smking (Spell check)

Line 114. Prevalence was 23.1% (Cross-check)

Line 113 to line 148. Written hurriedly. It would be appropriate if you could include table for quantitative data as well in order to gain clear understanding of the results and inferences obtained. 

Discussion: Line 172. Prevalence rate 23.1% (Cross-check)

Material and methods: Line 227. Please re-confirm if DNA concentration is 100 nM or 100 ng. Similarly cross check concentrations of primer and probes once again. 

Table 3. Probe (5'-3?)...... (should be 5'-3')

Author Response

Response to Reviewer 4 comments

The study aims to evaluate the prevalence of S. aureus and MRSA mec-A gene in the throat swabs from randomly selected Non-hospitalized population from Central Italy. New data generated seem to be just adequate.

Some of the suggestions are

Point 1: Italicize the bacterial names through out the manuscript.

Response 1: Authors agree with the Reviewer. Italics formatting was lost at some stage in the manuscript uploading process. The revised version has been corrected.

Point 2: Title: Suggest to re-frame the title. Avoid using bacterial name at the start of title.

Response 2: In response to the Reviewer’s suggestion, the title was revised: “Prevalence of Staphylococcus aureus and mec-A cassette in the throat of non-hospitalized individuals randomly selected in Central Italy“

Point 3: Abstract: Prevalence of S. aureus mentioned as 23.1% (Please cross check, should be 21.3%).

Response 3: Authors thank the Reviewer for the attention put in the revision process and for the detection of incongruency between the number of positive samples mentioned in text and the S. aureus prevalence reported in the original manuscript. The reported prevalence was correct (23.1%), but the number of positive samples mentioned in the text (184) was not. As reported in Table 2, the S. aureus positive samples were indeed 199 out of 861. The number of 184, wrongly reported before, was the number of samples positive for S. aureus but negative for mec-A.  The Abstract and the Results section were consequently revised.

Abstract (line 27): “The DNA of S. aureus was detected in 199 specimens (23.1%), while mec-A gene was detected in 27 samples (3.1%).”

Results (line 119): “The DNA of S. aureus was detected by real-time PCR in 199 specimens, while mec-A gene was detected in 27 samples.”

Point 4: Introduction: Line 75. S epidermidis and other coagulase-negative....(Write full bacterial name at its first appearance in the text)

Response 4: The manuscript was edited according to the Reviewer’s suggestion.

Point 5: Results: Table 1. Smking (Spell check)

Response 5: The typo was corrected

Point 6: Line 114. Prevalence was 23.1% (Cross-check)

Response 6: Please, see response 3.

Point 7: Line 113 to line 148. Written hurriedly. It would be appropriate if you could include table for quantitative data as well in order to gain clear understanding of the results and inferences obtained.

Response 7: The Results section was written avoiding any possible comment on purpose. This might explain the reviewer's perception of haste. On the other hand, any piece of data obtained in the investigation was explained and commented in the Discussion section.

To gain understanding of the results, the revised manuscript includes an additional table (Table 3). This table shows the occurrence of S. aureus and mec-A stratified by sex, smoking habit, and alcohol consumption.

As regard of quantitative data, the Authors thought that graphs would do a better job than tables. However, they felt that any graphs accompanying the correlation analysis would not significantly improve the understanding of the data.

Point 8: Discussion: Line 172. Prevalence rate 23.1% (Cross-check)

Response 8: Please see response 3

Point 9: Material and methods: Line 227. Please re-confirm if DNA concentration is 100 nM or 100 ng. Similarly cross check concentrations of primer and probes once again.

Response 9: DNA amount is now correctly stated as 100 ng, while the primer and probe concentrations were already right.

Point 10: Table 3. Probe (5'-3?)...... (should be 5'-3')

Response 10: The typo has been corrected (now Table 4)

Reviewer 5 Report

Dear Authors,

The present investigation explored the prevalence of mec-A and S. aureus sequences in throat swabs from randomly selected people from central Italy.

The study is of scientific interest and in line with the aims of the journal, however the manuscript should be copyedited by a native English speaker or copyediting service.  

The Material and Methods, Result, and Discussion sections were well described.

In my opinion, the manuscript is suitable for publication in this Journal.

Materials and Methods 

- “The human subjects included in the present study were randomly selected among 206 dental clinics of the University of Tor Vergata (Italy) and from the University of L’Aquila 207 (Italy)”. When patients were recruited?

Discussion

I suggest beginning the Discussion Section reporting and discussing the systemic consequences of S. aureus infection.

Author Response

Response to Reviewer 5 comments

Dear Authors,

The present investigation explored the prevalence of mec-A and S. aureus sequences in throat swabs from randomly selected people from central Italy.

The study is of scientific interest and in line with the aims of the journal, however the manuscript should be copyedited by a native English speaker or copyediting service. 

The Material and Methods, Result, and Discussion sections were well described.

In my opinion, the manuscript is suitable for publication in this Journal.

Point 1: Materials and Methods

- “The human subjects included in the present study were randomly selected among  dental clinics of the University of Tor Vergata (Italy) and from the University of L’Aquila (Italy)”. When patients were recruited?

Response 1: Authors added the time specification in the Mat&Met section, line 222: “The human subjects included in the present study were randomly selected among dental clinics of the University of Tor Vergata (Italy) and from the University of L’Aquila (Italy) between December 2017 and March 2019”

Point 2: Discussion

I suggest beginning the Discussion Section reporting and discussing the systemic consequences of S. aureus infection.

Response 2: The Authors agree with the Reviewer’s suggestion and have included the following paragraph to the Discussion section: “S. aureus is a gram-positive bacterium that colonizes the skin and mucosae in about 25–30% of healthy people. However, S. aureus can cause a range of diseases if it crosses the epithelial barrier and accesses to underlying tissues. This major human pathogen is a common cause of systemic infections such as infective endocarditis, osteomyelitis, epiglottitis, and sinus infections amongst others. Such infections can be life-threatening when uncured or in case of multi-drug resistance that can be acquired through mutational changes or via horizontal gene transfer by other bacteria (PMID: 16420592).“

Round 2

Reviewer 1 Report

No further comment

Reviewer 2 Report

The authors have improved their manuscript by respecting the suggestions and comments made!